# FEW HEADS ARE ENOUGH

## ABSTRACT

The costly self-attention layers in modern Transformers require memory and compute quadratic in sequence length. Existing approximation methods usually underperform and fail to obtain significant speedups in practice. Here we present Expert Projection Attention (EPA)—a novel method that reduces both compute and memory requirements and achieves wall-clock speedup, while matching the language modeling performance of baseline Transformers with the same parameter budget. EPA uses Mixture-of-Experts (MoE) layers for the value and output projections and requires 4 to 8 times fewer attention matrices than standard Transformers. Our novel attention can also be combined with MoE MLP layers, resulting in an efficient "Fast Transformer."[1]

## 1 INTRODUCTION

Large language models (LLMs) have demonstrated remarkable abilities (Radford et al., 2019; Brown et al., 2020; OpenAI, 2022; 2023) and incredible versatility (Bubeck et al., 2023). However, training enormous Transformers (Vaswani et al., 2017; Schmidhuber, 1992) necessitates a compute and memory budget that is well above what is available to most researchers, academic institutions, and even companies. In fact, even running them in inference mode, where the requirements are much weaker, requires a huge engineering effort (Gerganov, 2023). Thus, smaller but more capable models have also received significant attention (Touvron et al., 2023; Taori et al., 2023; Chiang et al., 2023; MistralAI, 2023; Stanić et al., 2023). However, even with these cutting-edge techniques, LLM training is beyond the reach of most researchers.

Recently, Csordás et al. (2023) have proposed to use a new non-competitive Mixture of Experts (MoE) model to accelerate Transformer training. The authors have shown that it performs on par with or can even outperform their parameter-matched dense counterparts with a fraction of the resource requirements. Previously in the literature, MoE models have been successfully used to scale Transformers to a very large number of parameters (Shazeer et al., 2017; Lewis et al., 2021; Lepikhin et al., 2021; Fedus et al., 2022; Clark et al., 2022; Chi et al., 2022), but without paying attention to their *parameter efficiency*. Importantly, all of these methods focus on the MLP layer, and not on the attention.

However, the attention layer (Schmidhuber, 1991; Bahdanau et al., 2015) in Transformers accounts for a significant proportion of both their compute and memory usage, especially for long context sizes. Linear attention (Schmidhuber, 1991; Katharopoulos et al., 2020; Choromanski et al., 2021; Schlag et al., 2021) was proposed as a remedy, but in practice, most methods fail to achieve significant speedups (Dao et al., 2022) and sometimes underperform compared to the exact attention.

As an alternative, MoE-based attention has been proposed (Zhang et al., 2022; Peng et al., 2020). However, they only achieve a modest reduction in computing and memory requirements, and typically require a lot of engineering tricks for successful training. Generally, MoE-based attention remains underexplored.

In this paper, we propose a novel MoE-based attention mechanism, called Expert Projection Attention (EPA), that aims to minimize the number of attention matrices required to be computed and stored. Our method is based on the $\sigma$-MoE by Csordás et al. (2023) and does not require regularization or additional tricks for stable training. Our method is capable of achieving predictive performance on par with parameter-matched baselines, with a fraction of the required compute and

---

[1]Here we will add a link to our public GitHub code repository upon acceptance.

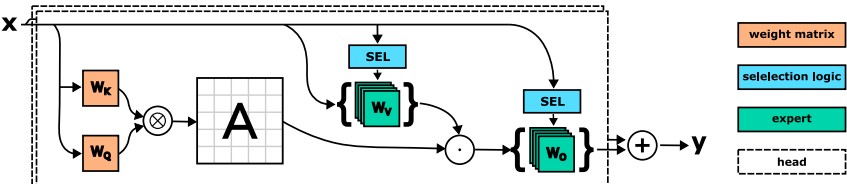

Figure 1: Schematic representation of EPA. It consists of few independent heads, each with multiple experts for value and output projections. Each head has a single attention matrix.

memory budget. We demonstrate this on a wide range of language modeling datasets and on two model sizes. We also show that models combining a $\sigma$-MoE-based MLP layer with our attention typically outperform dense baselines with identical parameter budgets, achieving a "Fast Transformer" Transformer model. Finally, we analyze the attention maps of our Expert Projection Attention, and show that the maximum of attention maps taken over all heads are qualitatively similar to the dense baselines, showing a significant reduction in redundancy without a loss of expressivity. Also, expert selections are often interpretable.

## 2 METHOD

### 2.1 BACKGROUND

The standard multi-head self-attention (MHA) layer (Vaswani et al., 2017) consists of four major steps: (1) computing key (K), query (Q), and value (V) projections, (2) computing the attention matrix, (3) using the attention matrix to project the values, and (4) mapping the projected values to the output. Let $h$, $T$, $d_{\text{model}}$, $d_{\text{head}}$ denote positive integers. Let $x \in \mathbb{R}^{T \times d_{\text{model}}}$ denote an input to the MHA layer, $T$ be the sequence length, and $d_{\text{model}}$ denote the size of the hidden representations of the model. $W_{\{K,V,Q\}}^{h} \in \mathbb{R}^{d_{\text{model}} \times d_{\text{head}}}$ are the projection matrices for head $h$. Then $K^{h} = xW_{K}^{h}$, $Q^{h} = xW_{Q}^{h}$, and $V^{h} = xW_{V}^{h}$ (thus $K^{h}, Q^{h}, V^{h} \in \mathbb{R}^{T \times d_{\text{head}}}$) are the keys, queries, and values, respectively. The attention matrix for the head $h$, $A^{h} \in \mathbb{R}^{T \times T}$, and the output $y \in \mathbb{R}^{T \times d_{\text{model}}}$ are calculated as follows:

$$A^{h} = \text{softmax}\left(\frac{1}{\sqrt{d_{\text{model}}}} Q^{h} K^{h\mathsf{T}}\right) \tag{1}$$

$$y = W_{O}(A^{0}V^{0}|A^{1}V^{1}|...|A^{H}V^{H}) \tag{2}$$

where $|$ denotes concatenation in the last dimension, the $\text{softmax}(\cdot)$ is also over the last dimension, and $W_{O} \in \mathbb{R}^{d_{\text{model}} \times H d_{\text{head}}}$. However, an alternative formulation reflects the role of $W_{O}$ better. Let us divide $W_{O}$ along the second dimension into submatrices for each head, $W_{O}^{h} \in \mathbb{R}^{d_{\text{model}} \times d_{\text{head}}}$, such that $W_{O} = W_{O}^{0}|W_{O}^{1}|...|W_{O}^{H}$. In this case, the output can be equivalently written as:

$$y = \sum_{h} W_{O}^{h} A^{h} V^{h} \tag{3}$$

From this, it can be seen that all computations are local to the heads. Computing the attention matrix $A^{h}$ and the readout $A^{h}V^{h}$ requires compute in order of $O(Hd_{\text{head}}T^{2})$ MACs (multiplication-accumulation operation). During training, it requires the storage of $O(HT^{2})$ for the attention matrices and $O(HTd_{\text{head}})$ numbers for storing the sub-results of the projections. Given a sufficiently long sequence, computing the attention matrix and projecting the values will dominate the compute requirements due to the quadratic dependence on the sequence length $T$.

### 2.2 FROM DENSE TO EXPERT PROJECTION ATTENTION

Our goal is to obtain resource reductions while maintaining the fundamental properties of attention and retaining a fully expressive attention matrix. In fact, there is still room for improvement: modern LLMs use tens of heads (Brown et al., 2020; Touvron et al., 2023). Are so many of them all necessary? As we show later in Sec. 3, indeed, naively reducing the number of heads (while keeping the same number of parameters by increasing the head dimension) results in performance loss.

Explaining the reason for the need for many heads is beyond the scope of this paper. Nevertheless, here are some hypotheses: (1) they provide multiple inputs for the operations that the network performs in each step, (2) they are specialized and provide inputs only for specific operations. In this case, each operation would use a different subset of heads. (3) They may also provide alternatives with different initializations, some being more successful than others, thus enabling better learning. Among these, some (2) and (3) offer an opportunity for resource savings: if not all heads are needed at the same time, it might be possible to switch between them. The simplest method of doing so is to produce a gating signal using a linear projection $\boldsymbol{W}_S \in \mathbb{R}^{d_{\text{model}} \times H}$, and use the ones with the highest activation, by replacing Eq. 3 with Eq. 6:

$$\boldsymbol{s} = \sigma\left(\boldsymbol{x}\boldsymbol{W}_S\right) \tag{4}$$

$$\mathcal{E} = \arg\text{topk}(\boldsymbol{s}, K), \mathcal{E} \subset \{1, ..., H\} \tag{5}$$

$$\boldsymbol{y}[t, c] = \sum_{h \in \mathcal{E}} \boldsymbol{s}[t, h](\boldsymbol{W}_O^h \boldsymbol{A}^h \boldsymbol{V}^h)[t, c] \tag{6}$$

where $\boldsymbol{y}[t, c]$ denotes indexing the specific element of the matrix, specifically denoting timestep $t$ and channel $c$. Following Csordás et al. (2023), we use a non-competitive selection function. Intuitively, this corresponds to choosing a subset of attention heads for each *output* position. Our preliminary experiments confirmed that this method is indeed feasible for language modeling on WikiText-103. However, it is difficult to achieve acceleration and memory savings with this method. To see why, notice that the entries of the attention matrix $\boldsymbol{A}^h$ depend on *pairs* of inputs in different positions, but the choice is made only based on the output position. Thus, in the worst case, all possible projections have to be computed on the "source side" for the keys and values, which we would like to avoid.

An alternative approach, which we propose here, is to perform the conditional computation on the projections, independently for the source side ($K$ and $V$) and the destination side ($Q$ and output). This avoids conditional computation that involves the attention matrix itself. The obvious way to make the projections conditional is to use Mixture of Experts (MoEs). In this case, the concepts of "heads" are not well defined anymore. Therefore, we define a head to be a specific, computed attention matrix. For each head $h$, we define a list of $E$ experts. Then, the projection matrices become $\boldsymbol{W}_K^{h,e}$, $\boldsymbol{W}_Q^{h,e}$, $\boldsymbol{W}_V^{h,e}$ and $\boldsymbol{W}_O^{h,e}$, where $h$ denotes the head index and $e$ the specific expert. Then we compute the source-side expert selection as following:

$$\boldsymbol{s}_S^h = \sigma(\boldsymbol{x}\boldsymbol{W}_S^h) \tag{7}$$

$$\mathcal{E}_S^h = \arg\text{topk}(\boldsymbol{s}_S^h, K), \mathcal{E}_S^h \subset \{1, ..., E\} \tag{8}$$

We compute the destination-side experts similarly: $\boldsymbol{s}_D^h = \sigma(\boldsymbol{x}\boldsymbol{W}_D^h)$, $\mathcal{E}_D^h = \arg\text{topk}(\boldsymbol{s}_D^h, K), \mathcal{E}_S^h \subset \{1, ..., E\}$. Then, the value projection $\boldsymbol{V}^h$ is computed as a weighted sum of the selected experts:

$$\boldsymbol{V}^h = \sum_{e \in \mathcal{E}_S^h} \boldsymbol{s}_S^h[e]\boldsymbol{x}\boldsymbol{W}_V^{h,e} \tag{9}$$

The key and query projections are computed similarly: $\boldsymbol{K}^h = \sum_{e \in \mathcal{E}_S^h} \boldsymbol{s}_S^h[e]\boldsymbol{x}\boldsymbol{W}_K^{h,e}$, and $\boldsymbol{Q}^h = \sum_{e \in \mathcal{E}_D^h} \boldsymbol{s}_D^h[e]\boldsymbol{x}\boldsymbol{W}_Q^{h,e}$. The output projection also becomes an MoE:

$$\boldsymbol{y} = \sum_{h=0}^{H-1} \sum_{e \in \mathcal{E}_D^h} \boldsymbol{W}_O^{h,e} \boldsymbol{A}^h \boldsymbol{V}^h \tag{10}$$

As we'll show, it is not necessary to make all projections MoEs. In Section 3.1 we show that keeping a single copy of the projections $Q$ and $K$ and reusing them for all experts is beneficial. We call this method Expert Projection Attention. If this method can reduce the number of heads $H$ by having more experts, $E$, then it provides an easy way to reduce the resource requirements of MHA. Note that our method does not depend on the specific implementation of the attention, allowing easy experimentation and research. A schematic representation is shown in Fig. 1.

Unlike standard MoE methods, we found that no regularization is necessary to achieve good performance with our method.

### 2.3 RESOURCE USAGE OF DIFFERENT METHODS

In this section, we discuss the compute and memory usage of different attention variants. We will define the compute in terms of the number of multiply-accumulate operations (MACs, also used by Zhang et al. (2022)), which is arguably better defined than FLOPs (e.g., does one step of the matrix multiplication count as 1 FLOP or 2? Do we include the softmax?). All calculations will be presented for a single attention layer for a single sequence, and they are presented this way in all our tables. Both the memory and compute requirements scale linearly with both the batch size and the number of layers.

Consider a sequence of inputs of length $T$, with representation size $d_{\text{model}}$. Let $d_{\text{head}}$ be the width of the $K, Q$, and $V$ projections used for the attention layer. For Transformer XL-style attention, let the size of the context be $CT$, where $C - 1$ is the number of past chunks included in the context of the current attention step. We can divide the computation into two major parts: calculating the projections, which do not involve the attention map, and calculating the attention map and projecting the sequence of values using it.

First, consider the case of the standard Transformer XL (Dai et al., 2019). Here, from the input $\boldsymbol{x} \in \mathbb{R}^{T \times d_{\text{model}}}$, we calculate the $\boldsymbol{K}^h, \boldsymbol{Q}^h, \boldsymbol{V}^h \in \mathbb{R}^{T \times d_{\text{head}}}$ using projection matrices of shape $\mathbb{R}^{d_{\text{model}} \times d_{\text{head}}}$. The output after the attention is projected in a similar manner (Eq. 3). Thus, the projections take a total of $4T d_{\text{model}} d_{\text{head}}$ MACs per head. For backpropagation, we have to store all the intermediate results. This takes $T d_{\text{head}}$ numbers of $\boldsymbol{K}^h, \boldsymbol{Q}^h$ and $\boldsymbol{V}^h$. Also, the projected values should be stored. They have an identical shape, therefore, the total memory used by projections is $4T d_{\text{head}}$ numbers per head. Now consider the resource usage related to the attention matrix. It involves calculating the product of $\boldsymbol{Q}^h \boldsymbol{K}^{h\mathsf{T}}$, which takes $d_{\text{head}} CT^2$ MACs (multiplication by $C$ is needed because the shape of $\boldsymbol{K}^h$ and $\boldsymbol{V}^h$ for Transformer XL is $CT \times d_{\text{head}}$). The projection of the values with the attention matrix $\boldsymbol{A}^h \boldsymbol{V}^h$ is similar. For the memory usage, the attention needs $CT^2$ numbers, but it needs to be stored both before and after the activation function. In addition, calculating the projection of the position encodings is necessary. This depends on the implementation, but in our case, it involves a matrix multiplication, and the total amount of computation is $2 d_{\text{head}} d_{\text{model}} TC$, and it needs $2 d_{\text{head}} TC$ numbers of storage. Thus the resource requirements are:

$$N_{\text{MAC}}^{\text{XL}} = H \left( 4T d_{\text{head}} d_{\text{model}} + 2CT^2 d_{\text{head}} + 2CT d_{\text{head}} d_{\text{model}} \right) \tag{11}$$

$$N_{\text{mem}}^{\text{XL}} = H \left( 4T d_{\text{head}} + 2CT^2 + 2CT d_{\text{head}} \right) \tag{12}$$

The resource usage of Expert Projection Attention is different. First, the number of heads $H$ is significantly reduced, but $d_{\text{head}}$ is typically larger. Additionally, there are $K$ experts active at the same time. Here, we only consider the case where the value and outputs are experts, but $\boldsymbol{Q}^h$ and $\boldsymbol{K}^h$ are not (this version performs the best; see Sec. 3.1). Then, we have two projections that are identical with that of Transformer XL, and two MoE-based projections. These use $TK d_{\text{model}} d_{\text{head}}$ MACs to calculate the projection and another $TK d_{\text{head}}$ to calculate their weighted average. With a smart kernel implementation, memory usage is not affected by $K$, thus the formula remains the same as Eq. 12 (note, however, that $H$ and $d_{\text{head}}$ are very different in practice). The compute requirement can be calculated as:

$$N_{\text{MAC}}^{\text{EPA}} = H \left( 2T d_{\text{head}} d_{\text{model}} + 2TK d_{\text{head}}(d_{\text{model}} + 1) + 2CT^2 d_{\text{head}} + 2CT d_{\text{head}} d_{\text{model}} \right) \tag{13}$$

Additionally, the expert selection logic needs minimal additional resources, which can be ignored. Note that the comparison between the MACs of the standard (Eq. 11) and Expert Projection Attention (Eq. 13) depends on the exact values of the hyper-parameters. However, as we'll see in Sec. 3, in our typical configurations, EPA provides good predictive performance with significantly lower $H$ compared to the standard Transformer, resulting in reduced resource usage in the end.

## 3 EXPERIMENTS

Following Csordás et al. (2023) we conduct our experiments in a *parameter-matched* setting which better reflects the expressivity of language models (than the FLOPS-matched setting often used to evaluate MoEs). Without this constraint, with MoEs it is often possible to compensate for a weaker method by adding more experts. We use and adopt the CUDA kernel of Csordás et al. (2023) for our purposes. To match the number of parameters of different models, we follow a systematic procedure. First, we measure the parameter count of the dense Transformer, which serves as our

target. Then, for each method, we set the total number of experts (including between heads, $HE$ for Expert Projection Attention) to the same as the original number of heads. We increase the head projection size $d_{\text{head}}$ to the maximum that keeps the parameter count below our target. Because our CUDA kernel supports only $d_{\text{head}}$ with multiples of 4, this often remains below the parameter count of the baseline. For further compensation, we slightly increase $d_{\text{ff}}$ until we achieve a match that differs from our target with no more than 100k parameters but never exceeds it. We do not claim that this parameter-matching method is optimal, but we aim to have a consistent algorithm that does not require tuning, which is prohibitively expensive and would have to be done for each model separately. Detailed hyperparameters of all our models can be found in Sec. A.2 in the Appendix.

For all datasets except the character-level Enwik8 (Hutter, 2006), we use sub-word units (Sennrich et al., 2016; Schuster & Nakajima, 2012) obtained with a SentencePiece tokenizer (Kudo & Richardson, 2018) with a vocabulary size of 8k tokens. Unless otherwise noted, all models, including ours, are Transformer XL (Dai et al., 2019), with the context size being twice the size of the active/current chunk.

All models are trained for 100k batches. Some of the datasets we consider (C4 (Raffel et al., 2020), and peS2o (Soldaini & Lo, 2023)) are much larger. In this case, we train on the first $10^5 * T * N_{\text{batch}}$ tokens of the dataset.

## 3.1 WHICH PROJECTIONS REQUIRE AN MOE?

As discussed in Sec. 2.2, each linear projection (K, V, Q, O) can potentially be replaced by an MoE. Here we first check which projection benefits from such a replacement. As we target the parameter-matched setting, having experts where they are not necessary can have a negative effect. Since they use a significant part of the parameter budget, they can reduce the number of parameters available for the more useful parts of the model. Thus, we did a search over all possible combinations of expert versus fixed projections with two active heads and compared them to the parameter-matched baseline on Wikitext 103. Our models have 47M parameters. We also include a parameter-matched baseline with two heads, which serves as a lower bound for the performance. The results are shown in Tab. 1. It can be seen that the output projection is necessary to match the performance of the baseline. Having key and query experts seems to be unnecessary. In fact, without the output and value experts, they even underperform the dense baseline with $H = 2$ heads. The best-performing model is the one with experts for both value and output projections. We use this model variant for all the other experiments in this paper.

Table 1: The performance of EPA with $E = 5$ experts and $H = 2$ heads. Different projections are either experts or fixed for the given head. Parameter-matched baseline with $H = 10$ and $H = 2$ are shown. Models sorted by perplexity. 47M parameters models on Wikitext 103.

| Model | $n_{\text{heads}}$ | V expert | K expert | Q expert | O expert | Perplexity |
|---|---|---|---|---|---|---|
| EPA | 2 | Y | N | N | Y | 12.27 |
| EPA | 2 | N | N | N | Y | 12.30 |
| Transformer XL | 10 | - | - | - | - | 12.31 |
| EPA | 2 | N | Y | N | Y | 12.36 |
| EPA | 2 | Y | Y | N | Y | 12.37 |
| EPA | 2 | Y | N | Y | Y | 12.42 |
| EPA | 2 | Y | N | N | N | 12.45 |
| EPA | 2 | N | N | Y | Y | 12.45 |
| EPA | 2 | Y | N | Y | N | 12.51 |
| EPA | 2 | Y | Y | Y | Y | 12.57 |
| EPA | 2 | N | Y | Y | Y | 12.59 |
| EPA | 2 | Y | Y | Y | N | 12.61 |
| EPA | 2 | Y | Y | N | N | 12.69 |
| Transformer XL | 2 | - | - | - | - | 12.74 |
| EPA | 2 | N | N | Y | N | 12.75 |
| EPA | 2 | N | Y | N | N | 12.79 |
| EPA | 2 | N | Y | Y | N | 12.90 |

## 3.2 COMPARING WITH MoA

The method most related to ours is the so-called Mixture of Attention Heads, or MoA (Zhang et al., 2022). They use a selection mechanism to choose active attention heads from a set of experts. However, they have a single set of $K$ and $V$ projections shared between experts; thus, acceleration is possible. However, in the original paper, the authors use a high number of selected heads (8-16) which seems necessary to achieve good performance. Thus, the resource reductions are moderate. Moreover, MoA uses three different regularizers, which have to be tuned independently.

We compare our method with our reimplementation of MoA with a different number of selected heads. Given the complexity of tuning its regularization coefficients, we take them directly from Zhang et al. (2022). For a fair comparison, we also integrated the non-competitive selection mechanism of Csordás et al. (2023) into MoA. The results are shown in Table 2. Similarly to our method, we found that with non-competitive selection, no regularization is required, and the predictive performance usually is superior to the original formulation. However, it still underperforms our method given a similar computation and memory budget.

Table 2: The performance of EPA compared to different MoA variants. MoA can outperform the baseline, but only at a price of using significantly more computing and memory. Also, EPA outperforms the baseline dense Transformer. Results are on Wikitext 103.

| Model | sel. mode | $n_{\text{heads}}$ | #params | Perplexity | MACs | Mem (floats) |
|---|---|---|---|---|---|---|
| MoA | sigmoid | 8 | 47M | 12.13 | 390.2M | 2.6M |
| MoA | sigmoid | 6 | 47M | 12.16 | 306.8M | 1.9M |
| EPA | sigmoid | 2 | 47M | 12.27 | 170.4M | 0.8M |
| Transformer XL | - | 10 | 47M | 12.31 | 453.4M | 3.5M |
| MoA | sigmoid | 4 | 47M | 12.39 | 223.5M | 1.3M |
| MoA | softmax | 4 | 47M | 12.60 | 223.5M | 1.3M |
| MoA | softmax | 6 | 47M | 12.64 | 306.8M | 1.9M |
| MoA | sigmoid | 2 | 47M | 12.65 | 140.1M | 0.7M |
| MoA | softmax | 8 | 47M | 12.77 | 390.2M | 2.6M |
| MoA | softmax | 2 | 47M | 12.84 | 140.1M | 0.7M |
| MoA | softmax | 8 | 262M | 9.50 | 2.9G | 9.9M |
| EPA | sigmoid | 2 | 262M | 9.55 | 2.0G | 2.9M |
| MoA | sigmoid | 8 | 262M | 9.56 | 2.9G | 9.9M |
| MoA | sigmoid | 12 | 262M | 9.58 | 4.1G | 14.7M |
| Transformer XL | - | 16 | 262M | 9.66 | 5.4G | 21.0M |
| MoA | softmax | 12 | 262M | 9.68 | 4.1G | 14.7M |
| MoA | softmax | 4 | 262M | 9.69 | 1.7G | 5.1M |
| MoA | sigmoid | 4 | 262M | 9.77 | 1.7G | 5.1M |
| MoA | softmax | 2 | 262M | 9.87 | 1.1G | 2.7M |
| MoA | sigmoid | 2 | 262M | 10.02 | 1.1G | 2.7M |

## 3.3 PERFORMANCE ON DIFFERENT DATASETS

We test our methods on a diverse set of language modeling datasets, including C4 (Raffel et al., 2020), Enwik8 (Hutter, 2006), peS2o (Soldaini & Lo, 2023), at two different scales: a 47M and a 262M parameters. The results are shown in Tab. 3. We compare our models to two baselines: one with the same number of heads as the total number of experts ($H \cdot E$) of the EPA models, and the other has the same number of heads as the number of active attention matrices ($H$) as our models. Our models always closely match the performance of the full, many-head baseline with the fraction of memory and compute requirements. Importantly, our method also achieves a wall-clock speedup, enough to accelerate the entire training pipeline by a factor of around 1.5 (see Appendix A.4 for more details). This confirms the competitiveness of our method.

### 3.3.1 FAST TRANSFORMER

The goal of achieving more resource-efficient Transformers includes reducing the resource requirements of both the MLP and the attention layers. Csordás et al. (2023) proposed a parameter-efficient

Table 3: The performance of EPA compared to baselines on different datasets with different model sizes. It can be seen that the predictive performance of our Expert Projection Attention model is comparable to the baselines, and is always better than the baseline with an equal number of heads. Perplexity is shown for Wikitext 103, C4 and peS2o datasets, and bits/character (bpc) for Enwik8.

| Model | Dataset | $n_{heads}$ | #params | ppl/bpc | MACs | Mem (floats) |
|---|---|---|---|---|---|---|
| EPA | C4 | 2 | 47M | 22.55 | 202.5M | 0.8M |
| Transformer XL | C4 | 10 | 47M | 22.62 | 453.4M | 3.5M |
| Transformer XL | C4 | 2 | 47M | 23.38 | 453.4M | 1.4M |
| EPA | C4 | 4 | 262M | 16.27 | 2.4G | 5.6M |
| Transformer XL | C4 | 16 | 262M | 16.41 | 5.4G | 21.0M |
| EPA | Wikitext 103 | 2 | 47M | 12.31 | 170.4M | 0.8M |
| Transformer XL | Wikitext 103 | 10 | 47M | 12.32 | 453.4M | 3.5M |
| Transformer XL | Wikitext 103 | 2 | 47M | 12.73 | 453.4M | 1.4M |
| EPA | Wikitext 103 | 2 | 262M | 9.77 | 2.0G | 2.9M |
| Transformer XL | Wikitext 103 | 16 | 262M | 9.82 | 5.4G | 21.0M |
| Transformer XL | Wikitext 103 | 2 | 262M | 10.09 | 5.4G | 6.3M |
| EPA | peS2o | 2 | 47M | 12.86 | 202.5M | 0.8M |
| Transformer XL | peS2o | 2 | 47M | 13.28 | 453.4M | 1.4M |
| Transformer XL | peS2o | 10 | 47M | 14.28 | 453.4M | 3.5M |
| Transformer XL | peS2o | 16 | 262M | 10.78 | 5.4G | 21.0M |
| EPA | peS2o | 4 | 262M | 10.81 | 2.4G | 5.6M |
| EPA | Enwik8 | 2 | 41M | 1.10 | 709.3M | 2.8M |
| Transformer XL | Enwik8 | 8 | 41M | 1.10 | 1.6G | 10.5M |
| Transformer XL | Enwik8 | 2 | 41M | 1.13 | 1.6G | 4.2M |

MoE method to accelerate the MLP layers. However, it remains unclear whether it can be efficiently combined with our Expert Projection Attention, or can have some negative interaction effect if combined in a "Fast Transformer", where every layer is MoE-based.

In order to verify this, we take the architecture proposed by Csordás et al. (2023) without any hyperparameter change and replace the attention layer with EPA. The hyperparameters for the attention are directly taken from the experiments shown in Tab. 3. The results are shown in Tab. 4. The combined, fully-MoE model often outperforms the dense baselines for each dataset and model size considered, except in the case of the 259M parameter model on the C4 dataset.

Table 4: The performance of Fast Transformer (Expert Projection Attention + $\sigma$-MoE (Csordás et al., 2023)) compared to baselines on different datasets and model sizes. Our Fast Transformer model is close or better compared to the baselines.

| Model | Dataset | $n_{heads}$ | #params | ppl/bpc | MACs | Mem (floats) |
|---|---|---|---|---|---|---|
| Fast Transformer | Wikitext 103 | 2 | 47M | 12.17 | 170.4M | 0.8M |
| Transformer XL | Wikitext 103 | 10 | 47M | 12.32 | 453.4M | 3.5M |
| Fast Transformer | Wikitext 103 | 4 | 259M | 9.81 | 2.4G | 5.6M |
| Transformer XL | Wikitext 103 | 16 | 262M | 9.85 | 5.4G | 21.0M |
| Fast Transformer | C4 | 2 | 47M | 22.09 | 202.5M | 0.8M |
| Transformer XL | C4 | 10 | 47M | 22.62 | 453.4M | 3.5M |
| Fast Transformer | C4 | 4 | 259M | 16.45 | 2.4G | 5.6M |
| Transformer XL | C4 | 16 | 262M | 17.85 | 5.4G | 21.0M |
| Fast Transformer | peS2o | 2 | 47M | 12.56 | 202.5M | 0.8M |
| Transformer XL | peS2o | 10 | 47M | 14.28 | 453.4M | 3.5M |
| Fast Transformer | peS2o | 4 | 259M | 9.86 | 2.4G | 5.6M |
| Transformer XL | peS2o | 16 | 262M | 10.83 | 5.4G | 21.0M |

## 4 RoPE Positional Encodings

All of our experiments so far have used a Transformer XL model. Thus, it remains unclear whether Expert Projection Attention is specific to this model or can be also used with other attention methods. As an alternative, we consider RoPE positional encodings Su et al. (2021) without the XL cache (thus, the attention matrices are square). We test these models on Wikitext 103. The results are shown in Tab. 5. Our method also performs well in this case.

Table 5: The performance of Expert Projection Attention compared to dense baseline on Wikitext 103, using RoPE positional encoding instead of Transformer XL.

| Model | Dataset | $n_{\text{heads}}$ | #params | ppl/bpc | MACs | Mem (floats) |
|---|---|---|---|---|---|---|
| EPA (RoPE) | Wikitext 103 | 2 | 45M | 12.75 | 285.6M | 1.3M |
| Transformer (RoPE) | Wikitext 103 | 10 | 45M | 12.78 | 560.9M | 6.1M |
| Transformer (RoPE) | Wikitext 103 | 2 | 45M | 12.96 | 560.9M | 1.9M |
| EPA (RoPE) | Wikitext 103 | 4 | 243M | 10.00 | 4.2G | 18.4M |
| Transformer (RoPE) | Wikitext 103 | 16 | 244M | 10.17 | 6.4G | 37.7M |
| Transformer (RoPE) | Wikitext 103 | 2 | 244M | 10.26 | 6.4G | 8.4M |

## 5 Analysis

In order to see how the network uses the attention heads, we trained a small, 6-layer, 8-head Transformer on ListOps (Nangia & Bowman, 2018; Csordás et al., 2022). The reason for this choice is that small, algorithmic tasks tend to be more interpretable compared to language models. We also train a parameter-matched, 2-head Expert Projection Attention model. Both models achieve around 95% accuracy on a held-out IID validation set, in contrast to the dense 2-head model, which saturates around 80%. Note that ListOps is a classification task and does not use autoregressive masking.

Following Csordás et al. (2022), we visualize the maximum of attention heads for each layer, both for the standard Transformer (Fig. 2a) and Expert Projection Attention (Fig. 2b). The attention maps are qualitatively similar. Note that the initialization and the learning dynamics are different for the two models, thus the overlap would not be perfect even with the same type of model. We show all the attention maps for both models in Fig. 4 and 3 in the Appendix.

In addition, we visualize individual attention heads for the Expert Projection Attention model. An example is shown in Fig. 2c. In addition to the attention map, we show the weight of the selected experts for both the value and output projection (denoted by V and O, respectively, on the sides of the attention map). Often it is possible to interpret the selection weights: here, the output experts specialize according to different operations, while the input ones distinguish numbers and closed parentheses. The attention map itself appears to distribute information about contiguous chunks of numbers. Similar plots for all heads are shown in Fig. 5 in the Appendix.

The attention maps of the language models are difficult to interpret. However, we visualized the attention maps of the 47M parameter Transformer XL and the Expert Projection Attention model from Tab. 3. We found them to be qualitatively similar. We also identified induction heads (Olsson et al., 2022) in both models, some examples shown for EPA in Fig. 6a and for Transformer in Fig. 6b in the appendix. Other typical vertical line-lined attention patterns are shown in Fig. 6c and 6d.

## 6 Related Work

The method most closely related to ours is MoA (Zhang et al., 2022), which introduces a MoE style attention. It defines each attention head as an expert but shares the key and value projections between them. Unlike in our case, each of the selected experts requires a separate attention matrix, which significantly increases its memory usage. Due to the use of a competitive softmax-based activation function in the selection network, it requires complex regularization to prevent expert collapse. In the original formulation, the number of active heads is high. We also confirmed in our experiments that MoA needs many attention heads to match the performance of the dense baseline (see Sec. 3.2), and it is only possible to do so with a significantly higher resource budget than our method.

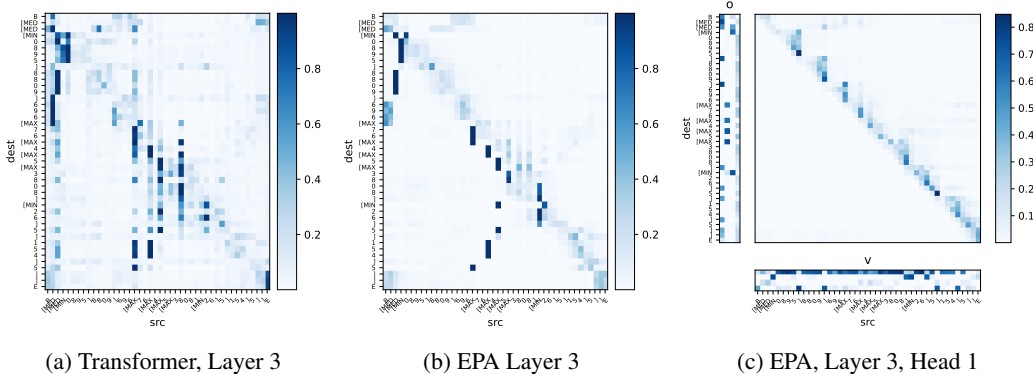

| (a) Transformer, Layer 3 | (b) EPA Layer 3 | (c) EPA, Layer 3, Head 1 |

Figure 2: An attention map of the (a) standard Transformer and (b) Expert Projection Attention. The maximum of all heads in the given layer are shown. (c) A head of EPA. On the left side of the attention plot, the selection weights of the output projection expert are shown. Similarly, at the bottom, the selection weights of the value experts are visible. In the selection maps, dark blue always corresponds to 1, while white is 0. The scale shown on the right is only for the attention.

Nguyen et al. (2022) analyze the attention matrices, and they conclude that they are usually low rank. Motivated by this, the authors construct a few (e.g., 2) "global attention matrices", and they compute each local matrix for specific heads by a weighted average of those. However, they average the logits, not the final matrix, so each individual head-specific matrix has to be computed. This means that in the best case, they can only save half of the computation associated with the attention matrix because the readout (Eq. 3) is still needed. For the same reason, memory savings are also low. The authors also use sampling of the attention matrices.

Peng et al. (2020) proposes to reweight the contribution of each head by a gating function. However, they only reduce the number of total attention heads by one, presumably to compensate for the parameters used by the selection logic. Their goal was not to reduce resource usage but to have better predictive performance, which they achieve. They use a softmax-based competitive selection mechanism. To avoid collapse, the gating function is trained only in some steps.

Csordás et al. (2023) introduce the non-competitive $\sigma$-MoE method that we also use for our attention mechanism. However, the authors focus on accelerating the MLPs and not the attention. More broadly, Shazeer et al. (2017) introduces sparsely-gated mixture of experts in LSTM (Hochreiter & Schmidhuber, 1997) networks. Fedus et al. (2021) introduces Mixture of Experts in Transformers. Lepikhin et al. (2021) trains a MoE-based LLM, and Clark et al. (2022) analyzes the scaling laws of MoE models. Lewis et al. (2021) introduces an alternative method for preventing collapse.

Dao et al. (2022) provides a hardware-aware CUDA implementation of the entire attention layer, which avoids storing the attention matrix. By saving memory bandwidth in this way, they achieve a significant wall clock time speedup, despite that the attention matrix should be recomputed in the backward pass. This is orthogonal to our method and they can be combined for further acceleration.

## 7 CONCLUSION

On a wide range of language modeling datasets with different model sizes, our novel Mixture-of-Experts-based attention method called Expert Projection Attention (EPA) achieves performance on par with parameter-matched dense counterparts, but with only a fraction of the computational cost and memory usage. EPA drastically reduces the number of attention matrices that have to be computed, by using MoE for the value and output projections. Our method is stable and does not need additional regularization to prevent degenerate solutions (a well-known practical issue in many existing MoE models). Our method can also be successfully combined with MoE MLP layers, to obtain a "Fast Transformer" where every layer is MoE-based, achieving a huge reduction in resource requirements.

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

## A  APPENDIX

### A.1  RESOURCE REQUIREMENTS OF MOA

The resource requirements of MoA (Peng et al., 2020) are very similar to those of Transformer XL (see Sec. 2.3 for more details), except that it uses a single shared $\boldsymbol{K}$ and $\boldsymbol{V}$ for each head.

$$N_{\text{MAC}}^{\text{MoA}} = (2H + 2)Td_{\text{head}}d_{\text{model}} + 2HCT^2 d_{\text{head}} + 2CTd_{\text{head}}d_{\text{model}} \tag{14}$$

$$N_{\text{mem}}^{\text{MoA}} = (2H + 2)Td_{\text{head}} + 2HCT^2 + 2CTd_{\text{head}} \tag{15}$$

### A.2  HYPERPARAMETERS

We train all our models with Adam optimizer (Kingma & Ba, 2015), with a batch size of 64, a learning rate of 0.00025, and gradient clipping with a maximum norm of $\kappa$. Large models ($> 200K$ parameters) use a learning rate warm-up of 4k steps. All models, except the Fast Transformer model, use a dropout on the MLP layers, $0.1$ for the small models and $0.2$ for the large ones. Detailed

hyperparameters are shown in the Tab. 6. $\sigma$-MoE related hyperparameters for the Fast Transformer models are identical to those of Csordás et al. (2023). For Transformer XL models, we always use a single additional chunk of context, both in training and validation time. $d_{\text{head}}$ and $d_{\text{ff}}$ are derived in a systematic way, see Sec. 3 for more details.

Table 6: Hyperparameters used for our models.

| Model | Dataset | $n_{\text{heads}}$ | #params | $d_{\text{head}}$ | $d_{\text{ff}}$ | E | K | T | $n_{\text{layers}}$ | $\kappa$ |
|---|---|---|---|---|---|---|---|---|---|---|
| EPA | C4 | 2 | 47M | 76 | 2080 | 5 | 3 | 256 | 16 | 0.1 |
| Transformer XL | C4 | 10 | 47M | 41 | 2053 | - | - | 256 | 16 | 0.1 |
| Transformer XL | C4 | 2 | 47M | 205 | 2053 | - | - | 256 | 16 | 0.1 |
| EPA | C4 | 4 | 262M | 112 | 4188 | 4 | 2 | 512 | 18 | 0.25 |
| Transformer XL | C4 | 16 | 262M | 64 | 4110 | - | - | 512 | 18 | 0.25 |
| EPA | Wikitext 103 | 2 | 47M | 76 | 2080 | 5 | 2 | 256 | 16 | 0.1 |
| Transformer XL | Wikitext 103 | 10 | 47M | 41 | 2053 | - | - | 256 | 16 | 0.1 |
| Transformer XL | Wikitext 103 | 2 | 47M | 205 | 2053 | - | - | 256 | 16 | 0.1 |
| EPA | Wikitext 103 | 2 | 262M | 132 | 4147 | 8 | 4 | 512 | 18 | 0.25 |
| Transformer XL | Wikitext 103 | 16 | 262M | 64 | 4110 | - | - | 512 | 18 | 0.25 |
| Transformer XL | Wikitext 103 | 2 | 262M | 512 | 4110 | - | - | 512 | 18 | 0.25 |
| EPA | peS2o | 2 | 47M | 76 | 2080 | 5 | 3 | 256 | 16 | 0.1 |
| Transformer XL | peS2o | 2 | 47M | 205 | 2053 | - | - | 256 | 16 | 0.1 |
| Transformer XL | peS2o | 10 | 47M | 41 | 2053 | - | - | 256 | 16 | 0.1 |
| Transformer XL | peS2o | 16 | 262M | 64 | 4110 | - | - | 512 | 18 | 0.25 |
| EPA | peS2o | 4 | 262M | 112 | 4188 | 4 | 2 | 512 | 18 | 0.25 |
| Transformer XL | Enwik8 | 8 | 41M | 64 | 2053 | - | - | 512 | 12 | 0.25 |
| EPA | Enwik8 | 2 | 41M | 112 | 2088 | 4 | 2 | 512 | 12 | 0.25 |
| EPA (RoPE) | Wikitext 103 | 2 | 45M | 64 | 2092 | 5 | 3 | 512 | 16 | 0.1 |
| Transformer (RoPE) | Wikitext 103 | 10 | 45M | 41 | 2053 | - | - | 512 | 16 | 0.1 |
| EPA (RoPE) | Wikitext 103 | 4 | 243M | 100 | 4136 | 4 | 2 | 1024 | 18 | 0.25 |
| Transformer (RoPE) | Wikitext 103 | 16 | 244M | 64 | 4110 | - | - | 1024 | 18 | 0.25 |
| Fast Transformer | Wikitext 103 | 2 | 47M | 76 | 1648 | 5 | 2 | 256 | 16 | 0.25 |
| Fast Transformer | Wikitext 103 | 4 | 272M | 128 | 4096 | 4 | 4 | 512 | 18 | 0.25 |
| Fast Transformer | C4 | 2 | 47M | 76 | 1648 | 5 | 3 | 256 | 16 | 0.25 |
| Fast Transformer | C4 | 4 | 272M | 128 | 4096 | 4 | 2 | 512 | 18 | 0.25 |
| Fast Transformer | peS2o | 2 | 47M | 76 | 1648 | 5 | 3 | 256 | 16 | 0.25 |

### A.2.1 A NOTE ON CHOOSING $n_{\text{HEADS}}$

Our preliminary experiments showed that a single head is usually not enough to match the performance of the baseline network, but two heads usually work well. Because of this, we always start by training a model with $n_{\text{heads}} = 2$ and increase it to $n_{\text{heads}} = 4$ if it does not match the performance of the baseline. We have not experimented with any other $n_{\text{heads}}$.

### A.3 A NOTE ON THE PARAMETER COUNT OF THE FAST TRANSFORMER

It can be seen in Tab. 4 that the parameter count of the Fast Transformer models is often less than that of their dense counterparts. The reason is that we normally compensate for the final difference in the number of parameters by increasing $d_{\text{ff}}$ (see Sec. 3 for details of the parameter matching). However, that can only be done in a very coarse-grained way with $\sigma$-MoE: the size of all experts must be increased at once, and the CUDA kernel supports only sizes of multiple of 4. Therefore, increasing the size of the experts would add too many parameters and the model would outgrow the baseline. For this reason, we simply keep the hyperparameters for Csordás et al. (2023) and combine them with our Expert Projection Attention configuration from Tab. 3.

## A.4 WALL-CLOCK TIME ESTIMATION

In all of our tables, we report the number of multiply-accumulate (MAC) operations following Zhang et al. (2022). The reason for this is that the actual wall-clock time is highly implementation and hardware-dependent. Nevertheless, we measured the runtime and total memory usage of our entire training pipeline (including the feedforward layer) to demonstrate that our current (suboptimal) implementation is already capable of providing wall-clock-time acceleration. We show the results in Tab. 7. The measurements are taken on identical hardware with the same implementation (including for the attention core), the only difference being the MoE-based projections for the attention. It can be seen that for both scales, our method trains around 1.5 times faster, while using 61%-67% as much memory as the baseline. Note that these measurements also include the MLP layers, the optimizer, and the gradient synchronization in the case of multi-GPU training.

Table 7: Real-world resource usage of our method. The numbers shown below are for training time for the whole pipeline, including the feedforward layers. It can be seen that EPA in the current implementation reduces both the runtime and the memory usage by a factor of 1.5-1.6.

| Model | Size | ms/iteration | Rel. iter. time | RAM/GPU | Rel. Mem. | #GPUs | GPU type |
|---|---|---|---|---|---|---|---|
| Trafo. XL | 47M | 770ms/iter | 1.0 | 20G | 1.0 | 1 | RTX 3090 |
| EPA | | 462ms/iter | **0.6** | 13.5G | **0.67** | | |
| Trafo. XL | 262M | 670ms/iter | 1.0 | 20.5G | 1.0 | 8 | V100 |
| EPA | | 442ms/iter | **0.65** | 12.5G | **0.61** | | |

## A.5 VISALIZING ALL ATTENTION HEADS

As discussed in Sec. 5, we analyze the attention maps of EPA and compare them with the dense models. We show all the attention maps of the models trained on ListOps in Fig. 3 and Fig. 3. We show individual heads of Expert Projection Attention, including the expert selection scores in Fig. 5. Some selected attention maps of our 47M parameter models on Wikitext 103 are shown in Fig. 6.

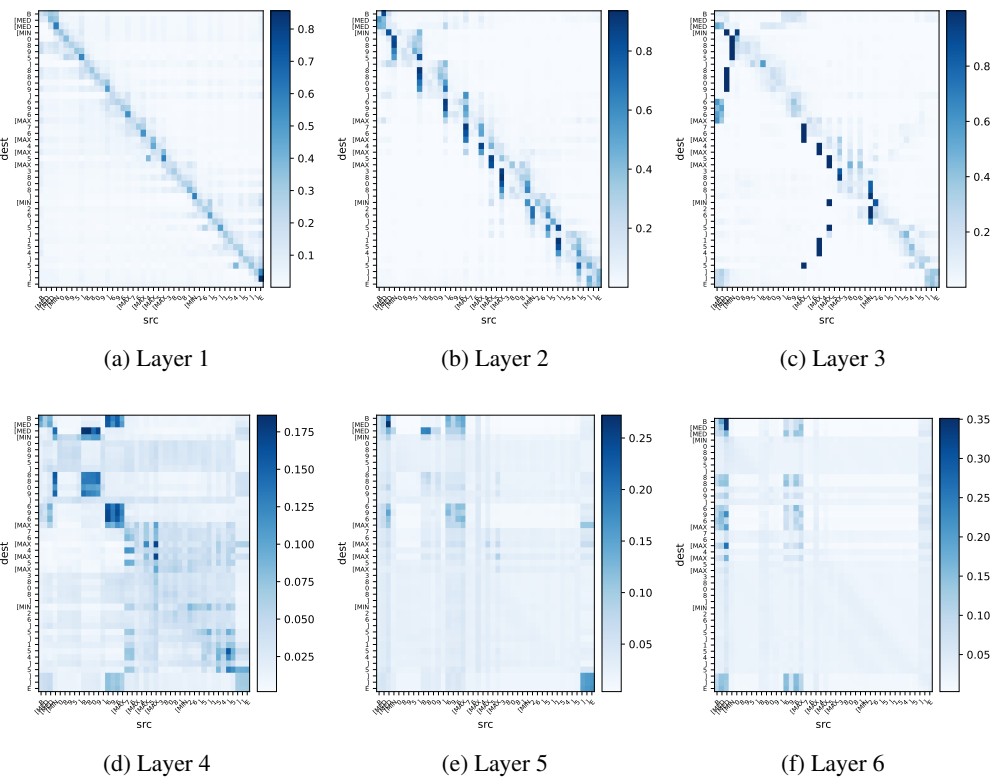

Figure 3: The maximum of all attention maps for a Expert Projection Attention model on ListOps.

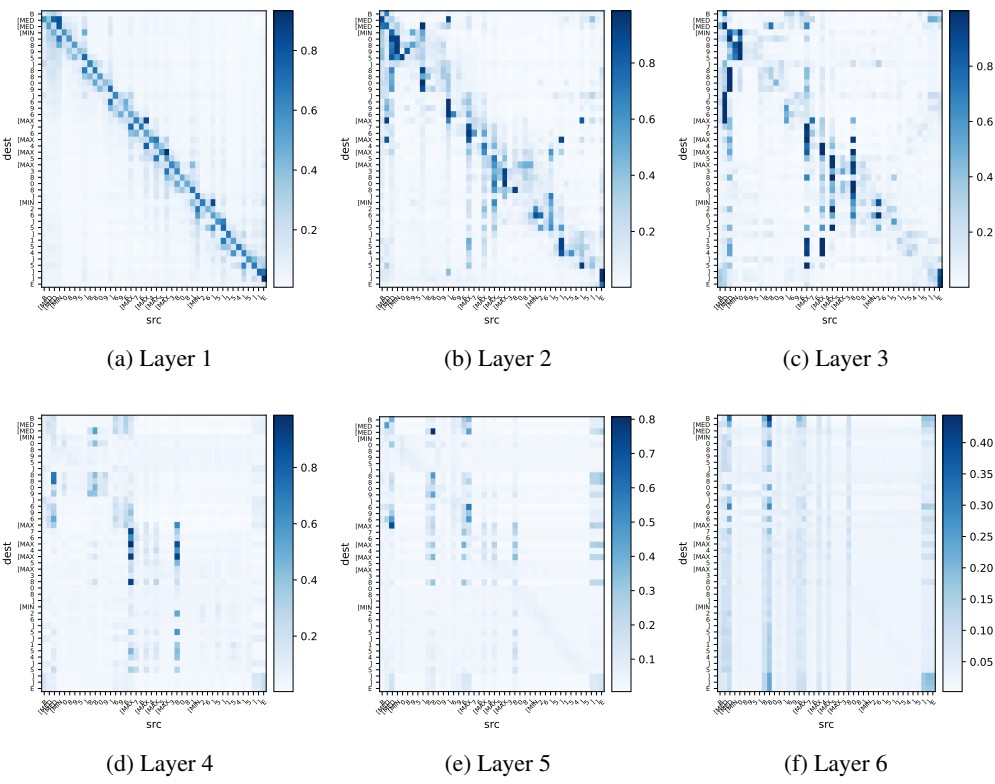

(a) Layer 1      (b) Layer 2      (c) Layer 3

(d) Layer 4      (e) Layer 5      (f) Layer 6

Figure 4: The maximum of all attention maps for a standard Transformer model on ListOps.

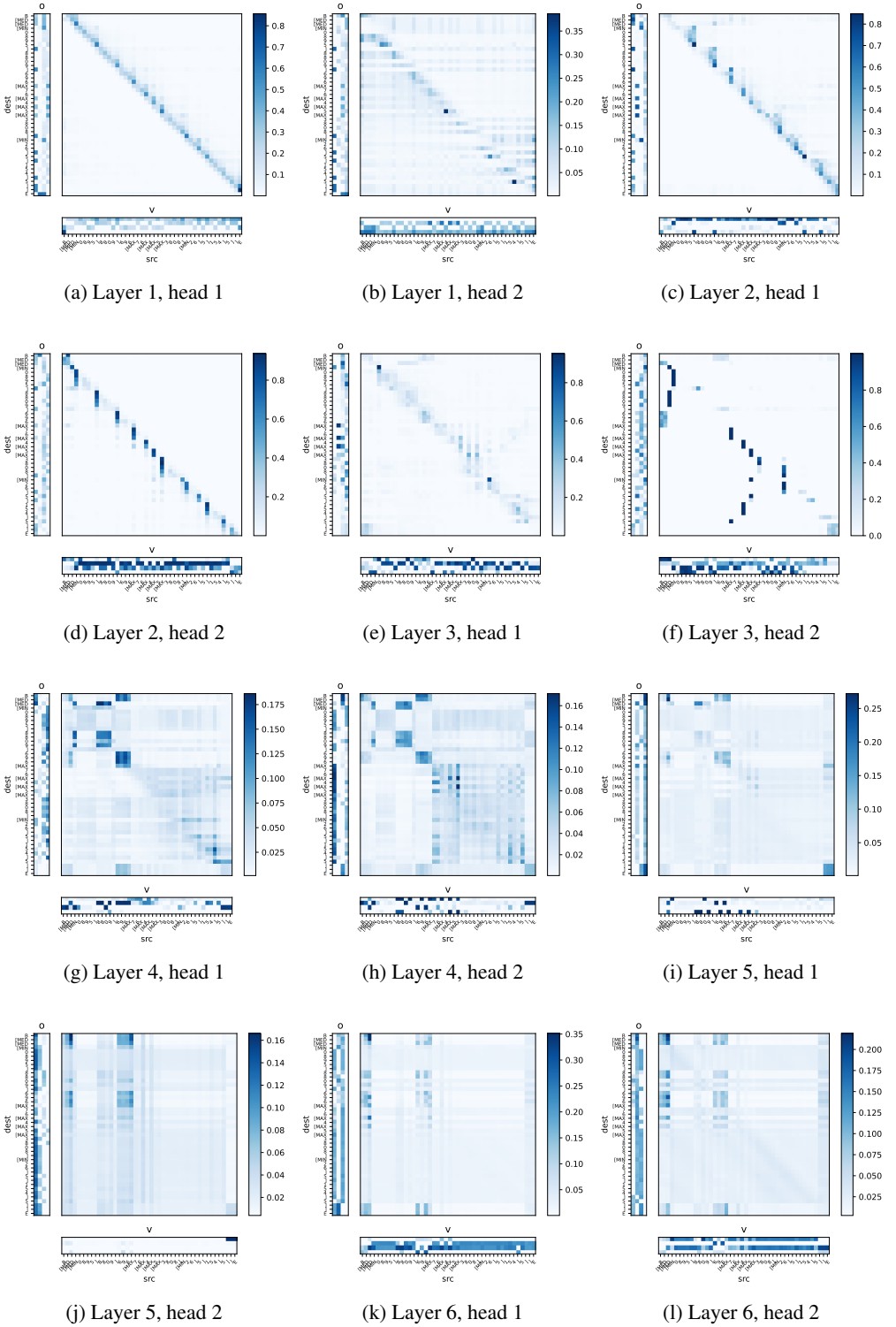

Figure 5: Details for individual heads of the EPA model on ListOps. On the left side of each attention plot, the selection of the output projection expert is shown. Similarly, at the bottom, the selection of the value projection selection is visible. In the selection maps, dark blue always corresponds to 1, while white is 0. The adaptive scale shown to the right of the attention map is for the map only.

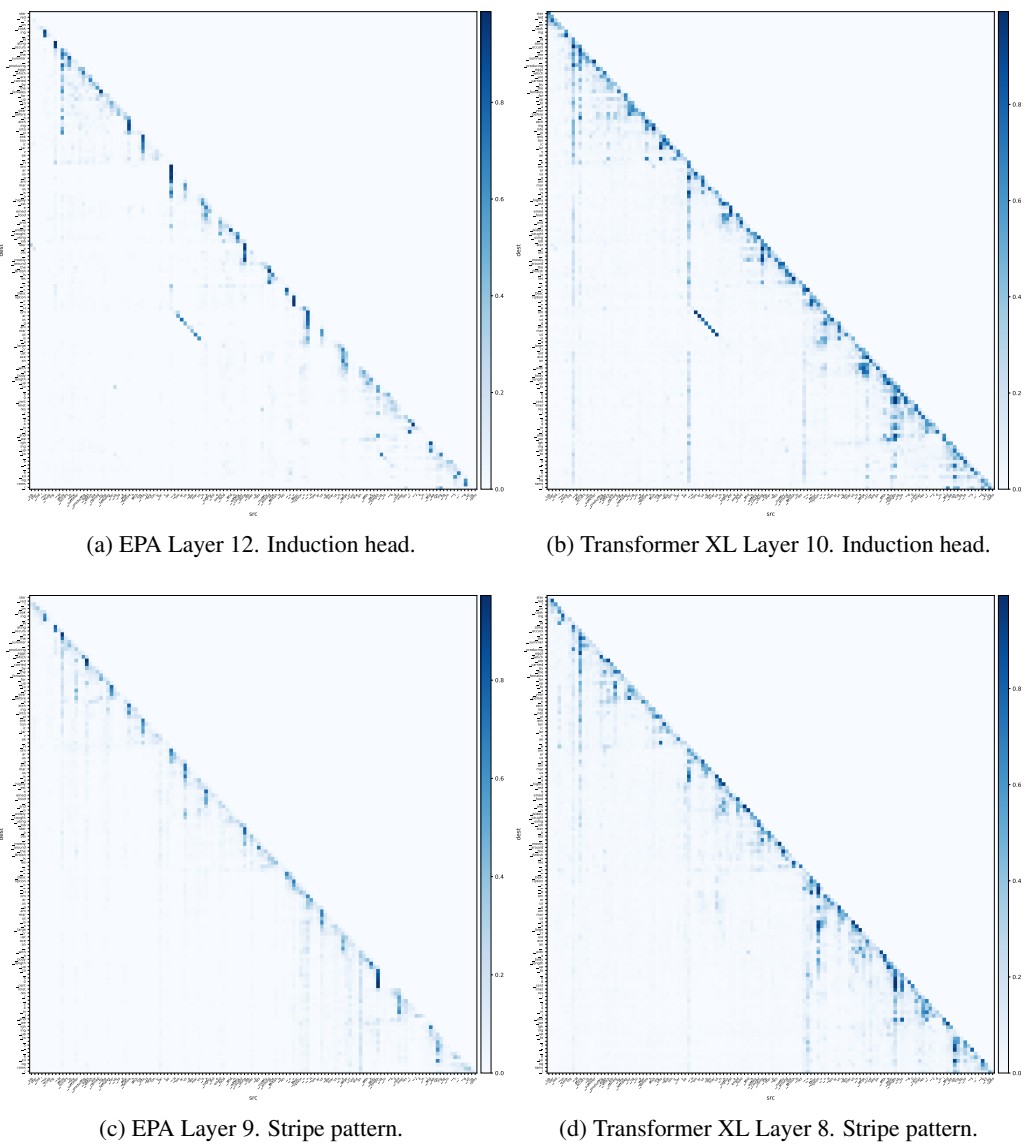

(a) EPA Layer 12. Induction head.      (b) Transformer XL Layer 10. Induction head.

(c) EPA Layer 9. Stripe pattern.      (d) Transformer XL Layer 8. Stripe pattern.

Figure 6: Induction head copying the rare name "Homarus" in (a) EPA and (b) Transformer XL baseline. The attention matrix is square because it is the first chunk of the sequence, without any extra context. Typical vertical line pattern in (c) EPA and (b) Transformer XL baseline.

