# OpenReview forum: "Few Heads are Enough"
_ICLR.cc/2024/Conference — Submitted to ICLR 2024_

### Official Review · Reviewer_dyhc · 2023-10-27

**Soundness:** 2 fair
**Presentation:** 2 fair
**Contribution:** 2 fair
**Rating:** 6
**Confidence:** 3

**Summary:**

This paper presents Expert Projection Attention (EPA). It reduces compute and memory requirement for attention by using MoE layers for values and output projections.

**Strengths:**

the method is clearly illustrated and the analysis in 2.3 is helpful for understanding the difference.

**Weaknesses:**

1. The author seems to misunderstand the position of flash-attention, see questions below.
2. the scale of experiment is small, how would this method generalize to larger models such as llama?
3. Some experiments have not been finished (Table 4).

**Questions:**

(1) In what sense flash-attention reduces compute. Do you mean FLOP or wall clock time? FA is exact attention and does not reduce FLOP, it is just a series of clever fusion. If it is wall clock time, then this paper should keep the definition consistent, and provide wall clock time analysis instead of MAC.
(2) "Unlike FlashAttention (Dao et al., 2022), it is research-friendly, because it does not hide the internal details of the attention mechanism inside a CUDA kernel." is an either arguably wrong or highly subjective judgement to flash-attention.

---

> ### Author Response · Authors · 2023-11-15
>
> We thank the reviewer for her/his valuable time reviewing our work. Please find our response as follows.
>
> ## Clarification on the weaknesses
>
> > 1. The author seems to misunderstand the position of flash-attention, see questions below.
>
> This is a false accusation (we do not understand what caused the reviewer think of this), as we clarify below.
>
> > 2. the scale of experiment is small, how would this method generalize to larger models such as llama?
>
> If the reviewer is asking us to run llama scale experiments, that is an unreasonable request.
> The cost of training llama-scale models is typically estimated as several millions of dollars. This is clearly not feasible for a university lab, nor necessary to show the core principle of an idea: in fact, we already had invested much compute for all these experiments; it is not as if we only reported on small/toy tasks such as Penn TreeBank. We report results on two scales, 47M and 262M parameters, and two different attention mechanisms (Transformer XL and RoPE) on four different datasets, some of them typically used for training much bigger models: Enwik8, Wikitext 103, C4 and peS2o. We think this is more than enough to demonstrate the feasibility of the model, and large companies with access to a sufficient amount of computing power can scale it up if interested.
>
> > 3. Some experiments have not been finished (Table 4).
>
> We are sorry about this (we had some technical problems with our cluster just before the deadline). The experiments have finished, and we updated Table 4 in the updated PDF.
> The results confirm that our model outperforms the baseline in every case.
>
> ## Response to the questions
>
> > (1) In what sense flash-attention reduces compute. Do you mean FLOP or wall clock time? FA is exact attention and does not reduce FLOP, it is just a series of clever fusion. If it is wall clock time, then this paper should keep the definition consistent, and provide wall clock time analysis instead of MAC.
>
> We report MACs by consistently following our main baseline paper (MoA, Zhang et al, 2023: Mixture of Attention Heads: Selecting Attention Heads Per Token). MAC is a good metric as it is not implementation-dependent, while the actual wall clock time is highly implementation-dependent. This having said, we suspect that what might have caused the confusion was a sentence in the abstract “...Flash-Attention reduces both compute…”. We admit that this formulation was confusing. We changed it to “...Flash-Attention reduces both run-time…”
>
> > (2) "Unlike FlashAttention (Dao et al., 2022), it is research-friendly, because it does not hide the internal details of the attention mechanism inside a CUDA kernel." is an either arguably wrong or highly subjective judgement to flash-attention.
>
> This is not at all wrong or subjective.  As we also clarified in our response to Reviewer DLPx (explaining why our method is “research friendly” when we also need a CUDA kernel), any modification to the “attention mechanism” in FlashAttention requires proficiency in GPU programming: this is unquestionable.
>
> For example, consider combining Transformer XL style relative positional encoding with FlashAttention. It requires loading positional encodings with different offsets for each row of the attention matrix, and combining it with the pre-computed query; constructing non-trivial memory access patterns that enable the reuse of positional encodings already loaded in shared memory… While all of this is doable, it is by no means trivial. Please check the Flash Attention GitHub repository (https://github.com/Dao-AILab/flash-attention) consists of 4500 lines of highly optimized CUDA code (an alternative Triton implementation around 1000 lines also exists).
> In contrast, our method can be trivially combined with PyTorch implementations of RoPE , Transformer XL style attention, Alibi, softmax1 activation, (and also with Flash Attention), etc.  None of these require any modification to our CUDA kernel, because the CUDA kernel is only used for the V and O projections (and optionally K and Q).
>
> ## General comments
>
> We respectfully note that the provided review is very brief and lacks specific comments/criticisms on our method. This makes it impossible for us to understand the justifications for the scores of “Soundness: 1 poor”, “Contribution: 1 poor” and a final score of 3 (while we acknowledge the rather low confidence score of 3). We emphasize that our EPA provides a novel method to significantly reduce both the compute (MACs) and memory of attention in a research-friendly way. As discussed above (under “Clarifications on the weaknesses”), we rigorously tested it across various scales, datasets, and attention variants with positive outcomes. In light of this, we kindly request more detailed feedback on our methodology, or specific areas of concern that led to the current evaluation, or please consider revising the scores. Thank you for your consideration.

---

> ### Comment · Reviewer_dyhc · 2023-11-15
> **Response to authors**
>
> Thank you for getting back to the reviews. The reviewer is happy to see all experiments are finished now.
>
> Given the current updates, the main question that affects whether the reviewer can raise the score is: Does the method achieve wall clock time speedup?
>
> MACs is not the correct measure, as reviewer obUv points out. Alternatively, if you don't think the current implementation leads to wall clock time speedup, you should give a reasonable implementation draft/plan that can lead to wall clock time speed up. If you do not have this plan in mind, you should change your position, and write something like "future work can improve on this, given the MAC is theoretically lower." If this is the case, you should reconsider why you are mentioning flash attention, the reviewer is happy to review again if you decide to reposition.

---

> > ### Author Response · Authors · 2023-11-17
> >
> > We thank the reviewer for his/her prompt reaction.
> >
> > > Given the current updates, the main question that affects whether the reviewer can raise the score is: Does the method achieve wall clock time speedup?
> >
> > Yes, it does. Please see indicative timing measurements we added in Appendix A.4. The wall clock speedup for the current implementation on both 47M and 262M scales is around 1.5-1.6 compared to the dense baseline, along with the memory reduction which is similar. We updated the PDF again to emphasize this acceleration better.
> >
> > We note that the current implementation still has room for improvements: async loads to shared memory, merging the weighted average calculation in the kernel, using bucket sort for preprocessing instead of torch sort, etc.
> >
> > We also would like to clarify one very important point regarding the goal of this work. The general goal of typical research on efficient attention is to make/train/evaluate Transformer models faster. Here we emphasize that such methods also have another important potential consequence in **accelerating research** on attention based models (by making them resource efficient). We consider this as an extremely important aspect today, potentially allowing for “resource-limited” organizations (e.g., in academia) to still conduct research on reasonably large models.
> > However, for a “fast attention” method to be effective in accelerating research, it also has to be “research friendly”: certain existing methods do not allow modifications of certain aspects of attention mechanisms, while others may be more flexible. Such flexibility is required for “research friendliness”.
> >
> > This is where our reference to FlashAttention becomes relevant. It is an example of a *non-flexible* acceleration method: it does not allow a researcher to modify the attention mechanism itself (which is hard-coded within their CUDA kernel), while our method still leaves such a possibility (since our CUDA kernel only replaces the linear projection layers to obtain query/key/values).
> >
> > For example, FlashAttention can **not** be used with the Transformer-XL baseline, or more generally some other positional encoding (e.g. XL-style relative positional encoding) or activation functions (e.g. geometric attention). In contrast, our method can be trivially combined with RoPE, Transformer XL style attention, Alibi, (and also with Flash Attention), etc. None of these require any modification to our CUDA kernel.

---

> > > ### Comment · Reviewer_dyhc · 2023-11-17
> > >
> > > Thank you for getting back to this important question. I have raised the score because the implementation has achieved wall clock time speedup, and has the potential to achieve more.

---

> ### Author Response · Authors · 2023-11-17
>
> Thank you very much for the score update. Now naturally, we'd like to know the reasons/justifications why the reviewer still votes for borderline reject.
>
> Based on our discussion above, it seems that we thoroughly addressed and resolved all the concerns the reviewer had initially raised. We would like to respond to any remaining concerns. Thank you.

---

> > ### Comment · Reviewer_dyhc · 2023-11-17
> >
> > Thanks for the question. This is a timely score update to reflect the fact that the important concern of wall clock time speedup has been addressed. I will go over all details again over tonight/weekend and summarize questions not yet been addressed.

---

> > ### Comment · Reviewer_dyhc · 2023-11-20
> > **Response to Authors**
> >
> > Hi there, thanks for the patience, the reviewer has re-read the paper again given the rebuttal, and some of the initial concerns are happily addressed.
> >
> > One remaining important question, I have been thinking for a while, and finally find out is the weird position of flash attention. Concretely:
> > (1) You are using flash-attention as a comparison from the very beginning of the paper, but your method is not comparing with it. If I understand correctly, your method has nothing to do with flash attention, and you could have started the paper with something else as motivations, such as MoA, or any MoE style attention. Motivating with flash attention seems highly irrelevant.
> > (2) You seem to have very negative wordings against flash-attention, concretely, e.g. "because it does not hide the internal details of the attention mechanism inside a CUDA kernel". You have put flash-attention in a very awkward position, and I don't know whether this is your intention.
> > (3) your accusation to flash-attention is still subjective, and I couldn't agree on your statement in last rebuttal, e.g. "it does not allow a researcher to modify the attention mechanism itself (which is hard-coded within their CUDA kernel), while our method still leaves such a possibility (since our CUDA kernel only replaces the linear projection layers to obtain query/key/values)."
> >
> > This is not a scientific statement, there is no measure on how hard it is to modify a codebase. The closest measure is probably LoC. You can say you sample 20 papers modified based on flash-attention, and how many LoC they need, with a comparison to your method. In addition, you seems to be very confident on the difficulty on "which requires proficiency in GPU programming". Firstly, I don't know how to judge this difficulty. Do you want to randomly sample x people in this field and count how many percent of them don't know GPU programming? Secondly, this is wrong. There are (1) other implementation of flash-attention, e.g. Triton, that allows modification as easy as near PyTorch Level. (2) work on modifying attention pattern that does not change the source code of flash attention, e.g. longnet[1].
> >
> > Again, the reviewer is trying to improve the paper, and have spent a lot of time on this. If you want to keep the flash attention flow, please be scientific. If you agree with my argument, please simply delete these sentences against flash attention. That will make the paper flow much better.
> >
> > [1] LONGNET: Scaling Transformers to 1,000,000,000 Tokens.

---

> > > ### Author Response · Authors · 2023-11-20
> > >
> > > We would like to thank the reviewer for his/her valuable time reviewing the paper. We appreciate his/her effort to improve the quality of the paper.
> > >
> > > We agree with the reviewer that the best thing to do here is to remove all mentions of FlashAttention (except in the related work section); this would remove all the sources of confusions discussed in this rebuttal from the beginning, without diminishing any of our core contributions.
> > >
> > > We thank the reviewer for pointing this out, and we think the current version of the paper better reflects the original goal of the method and causes less confusion. Please see the updated PDF (unfortunately we cannot change the Abstract on OpenReview, so the modifications are present only in the PDF).
> > >
> > >
> > > PS: From the beginning, our intention has never been to compete with Flash Attention or downplay it. There were in total 4 references to Flash Attention in the paper: 1. in the abstract as an example of a different type of fast attention method, 2. at the end of introduction, mentioning research friendliness, 3. at the end of P3, mentioning that it depends on the attention implementation,  and 4. in the related work section. There is one additional citation to it, but that is a reference for the sparsity-based methods typically not achieving wall time speedup, which they discuss in their paper. We removed mentions 1, 2, and 3 from the paper.
> > >
> > > Regarding the comment on research friendless (which has been removed), we had assumed it was a well-established consensus that modifying CUDA/Triton code is more challenging than replacing linear layers in PyTorch, but we agree with the reviewer that this is not based on any scientific metric.

---

> > > > ### Comment · Reviewer_dyhc · 2023-11-20
> > > > **Reply to Authors**
> > > >
> > > > I have looked over on the new paper. This is much better and clearly.
> > > >
> > > > Summary to AC: there are three main concerns (1) flash attention position, (2) wall clock time confusion versus MACs, (3) experiment scale not large enough like llama.
> > > >
> > > > The authors have addressed (1) and (2). I can agree on (3) that pretraining Llama scale is not practical, and given the number of dimension of experiments targeted on, I am fine with (3).
> > > >
> > > > I have raised my score to 6 to reflect this.
> > > >
> > > > Final suggestion to Author:
> > > >     In a strong paper, (3) should be addressed. In the future, you should consider how to design the method with fine-tuning paradigm. E.g, check how GQA paper positions this. There are ways to design new models, and reusing pretraining weights. In that case, your paper values will be much clearer.

---

> > > > > ### Author Response · Authors · 2023-11-21
> > > > >
> > > > > We would like to thank the reviewer for raising the score and for the very nice summary.
> > > > >
> > > > > Regarding point (3), we would like to emphasize that we agree with the reviewer that scale is becoming more and more important. However, we also note that our goal with this paper was to accelerate research to explore architectures that are different from standard Transformers. If the field discourages any new research that is different from “finetuning a pretrained model” we risk overfitting to the current state of the field even more than with the hardware lottery [1]. We would also like to note that finetuning a pretrained model to use EPA might be possible similarly to [2] or [3]. This was not among our original goals, but it could make an interesting followup work.
> > > > >
> > > > >
> > > > > 1. Sara Hooker, 2020: The Hardware Lottery
> > > > > 2. Ainslie et al, 2023: GQA: Training Generalized Multi-Query Transformer Models from Multi-Head Checkpoints
> > > > > 3. Zhang et al, 2021: MoEfication: Transformer Feed-forward Layers are Mixtures of Experts

---

> > > > > > ### Comment · Reviewer_dyhc · 2023-11-21
> > > > > >
> > > > > > thanks! good comment.

---

### Official Review · Reviewer_obUv · 2023-10-31

**Soundness:** 3 good
**Presentation:** 2 fair
**Contribution:** 2 fair
**Rating:** 5
**Confidence:** 4

**Summary:**

The authors propose a method for selecting one head but keeping a small number of Q,K and V matrix.
I really feel this paper is quite poorly written. First of all, the notations are extremely unclear especially in the method section. The authors have a schematic in
Figure 1 but it's unclear what this schematic means. There is no explanation of different boxes.

The authors constantly compare to FlashAttention, they have not even a single run time comparison.
I think the paper needs a complete re-write. The method section is using non-standard notation where is unclear what dimensions they are reducing to. The experiments do not talk about fine-tuning overheads, do not have a single timing results.

**Strengths:**

The idea at a high level looks decent. However, the poor writing and underwhelming evaluation really makes it hard to appreciate it.

**Weaknesses:**

Please see the summary.

**Questions:**

Please see the summary.

---

> ### Author Response · Authors · 2023-11-15
>
> We would like to thank the reviewer for his/her valuable time.
>
> We are surprised by the reviewer’s comment on the quality of writing (“the poor writing and underwhelming evaluation really makes it hard to appreciate it.”, “needs complete re-write”). In contrast, Reviewer DLPx highlights the clarity as one of the strengths of the paper in his/her high-quality review, and even Reviewer dyhc states that "the method is clearly illustrated". We’d like to hear more specific clarifications/suggestions on what confused the reviewer (including the comment regarding the notation).
>
> We would like to point out that the reviewer’s summary does not reflect what our algorithm does. We have a *few* heads (not 1), and the value and output projections (not “Q, K, and V”) are a mixture of experts.
>
> Regarding the timing results, we report the commonly used metric which is the number of “Multiply-Accumulated (MAC) operations” required for each method. We do this by following our main baseline paper (MoA, Zhang et al, 2023: Mixture of Attention Heads: Selecting Attention Heads Per Token). MAC is a good metric as it is not implementation-dependent. FlashAttention does not change the number of MAC operations, so it is the same as the baseline dense Transformer. If the reviewer is curious about the wall-clock time of our current implementation, we also measured the resource usage for both the baseline Transformer and EPA: please refer to Appendix 4 of our updated manuscript for more details. In short, EPA with the current (suboptimal) implementation is already capable of accelerating the entire training pipeline by a factor of 1.5-1.6, and reducing the total memory usage by about the same factor.
>
> We kindly ask the reviewer to clarify what “fine-tuning overheads” we should talk about. In fact, because our method accelerates the Transformer architecture, it accelerates both the training and fine-tuning phases.
>
> Overall, we find the current review very brief, and lacks technical comments/criticisms about our method. The reviewer rates our work as 3 (reject) with a high confidence score of 4; this implies responsibility. We respectfully ask for much more specific criticisms and justifications for such a rating.

---

> > ### Comment · Reviewer_obUv · 2023-11-15
> > **Clarifications.**
> >
> > First of all my apologies, I should have been more specific. I am sure different reviewers are entitled to their own opinion. I certainly believe you need to reposition this paper.  Let me be specific in my criticism.
> >
> > 1. Figure 1,  unclear what you mean. There is limited text to support. For the figure I will expect a block of text describing what it means.
> >
> > 2.  Criticism around notation - I followed your description till equation 10, then you say - " As we’ll show, it is not necessary to make all projections MoEs. In Section 3.1 we show that keeping
> > a single copy of the projections Q and K and reusing them for all experts is beneficial. We call this
> > method Expert Projection Attention", which to me means equation 10 might not be what actually you are using. Leaving me unable to understand what exactly do you mean and a proper definition.
> >
> > 3. Regarding MAC operations and not runtime as a metric - We have time and time seen MAC operations are not a substitute for actual runtime operation, whether it be pruning, quantization or compression In Appendix 4 you have not detailed what is the baseline implementation you are using. What codebase is it ? Without that those numbers mean little.
> >
> > 4. Fine-Tuning - My understanding is that you will need to perform some amount of fine-tuning before you can use your selected method, what is the overhead regarding this.
> >
> > 5. Finally - Other reviewers have made a point about comparison regarding Flash Attention and comparison with it. I have similar concerns.

---

> > > ### Author Response · Authors · 2023-11-17
> > > **Response to clarifications (part 1/2)**
> > >
> > > We thank the reviewer for his/her prompt response.
> > >
> > > > Figure 1, unclear what you mean. There is limited text to support. For the figure I will expect a block of text describing what it means.
> > >
> > > Figure 1 is a schematic depiction of the method that is described in Section 2.2. The meaning of individual boxes are shown on a legend on the right side of the figure: orange boxes are standard matrix multiplications with a weight matrix, blue boxes are the selection logic (corresponding to Eq. 7 and 8), the stack of green boxes represent the different experts (​​$W_{*}^{h,e}$). The figure shows 2 heads (the two dashed line boxes containing all the other boxes). The output of the heads are summed. The box with the grid and the big letter ‘A’ is the attention matrix.
> > >
> > > > Criticism around notation - I followed your description till equation 10, then you say - " As we’ll show, it is not necessary to make all projections MoEs. In Section 3.1 we show that keeping a single copy of the projections Q and K and reusing them for all experts is beneficial. We call this method Expert Projection Attention", which to me means equation 10 might not be what actually you are using. Leaving me unable to understand what exactly do you mean and a proper definition.
> > >
> > > We are glad to hear that everything is clear up to Eq. 10. We remind that an attention layer has 4 projection layers: three input projections (keys, queries and values) and one output projection. The model described in Sec. 2.2 (ending at Eq. 10.) corresponds to a model variation where we replace ALL/each of these 4 projections by an MoE. THEN, experimentally (Sec. 3.1), we show that not all of them actually has to be an MoE; the best model we found uses the regular linear projection for K and Q (i.e., mixture of expert Eqs for $K^h$ and $Q^h$ which are between Eqs. 9 and 10 are replaced by a simple linear projection). We hope this completely clarifies the reviewer’s confusion.
> > >
> > > Also, we note that we might not call this a “notation” problem (since the reviewer has successfully followed up to Eq 10).
> > >
> > > > Regarding MAC operations and not runtime as a metric - We have time and time seen MAC operations are not a substitute for actual runtime operation, whether it be pruning, quantization or compression
> > >
> > > Both the MAC operations and timings have their advantages and disadvantages: MACs are hardware and implementation-independent, but do not necessarily reflect in wall-clock speedup, while timing results are sensitive to the specific implementation. Our baseline paper (MoA, Zhang et al, 2023: Mixture of Attention Heads: Selecting Attention Heads Per Token) uses MACs as their main metric.
> > >
> > > Please note that we also added the runtimes relative to the dense baseline in Table 7 in Appendix 4 for easier comparison (see below).
> > >
> > >  > In Appendix 4 you have not detailed what is the baseline implementation you are using. What codebase is it ?
> > >
> > > We use our custom implementation in PyTorch, based on the publicly available Transformer XL code (https://github.com/kimiyoung/transformer-xl).
> > >
> > > Please note that knowing the codebase is not enough to make sense of absolute timing results because they depend on the hardware.
> > >
> > > > Without that those numbers mean little.
> > >
> > > They are meaningful in the sense that all models are tested in a controlled environment: with the *same hardware* and *same implementation* (including the attention core),  with the only difference being the presence of our modification or not. We updated the PDF again to further clarify and emphasize the wall-clock speedup and memory reductions to Table 7 in Appendix A.4.
> > >
> > > > My understanding is that you will need to perform some amount of fine-tuning before you can use your selected method, what is the overhead regarding this.
> > >
> > > Just to make sure: we remind that in the context of language modeling, “fine-tuning” refers to the process where we take a general “pre-trained model” and continue to train it (i.e., fine-tune it) on some downstream task. Our method accelerates the Transformer architecture: it will provide the same *acceleration* (as *opposed to overhead*) during fine-tuning as during the training.

---

> > > > ### Author Response · Authors · 2023-11-17
> > > > **Response to clarifications (part 2/2)**
> > > >
> > > > > Other reviewers have made a point about comparison regarding Flash Attention and comparison with it. I have similar concerns.
> > > >
> > > > We also would like to clarify one very important point regarding the goal of this work. The general goal of typical research on efficient attention is to make/train/evaluate Transformer models faster. Here we emphasize that such methods also have another important potential consequence in **accelerating research** on attention based models (by making them resource efficient). We consider this as an extremely important aspect today, potentially allowing for “resource-limited” organizations (e.g., in academia) to still conduct research on reasonably large models.
> > > > However, for a “fast attention” method to be effective in accelerating research, it also has to be “research friendly”: certain existing methods do not allow modifications of certain aspects of attention mechanisms, while others may be more flexible. Such flexibility is required for “research friendliness”.
> > > >
> > > > This is where our reference to FlashAttention becomes relevant. It is an example of a *non-flexible* acceleration method: it does not allow a researcher to modify the attention mechanism itself (which is hard-coded within their CUDA kernel), while our method still leaves such a possibility (since our CUDA kernel only replaces the linear projection layers to obtain query/key/values).
> > > >
> > > > For example, FlashAttention can **not** be used with the Transformer-XL baseline, or more generally some other positional encoding (e.g. XL-style relative positional encoding) or activation functions (e.g. geometric attention). In contrast, our method can be trivially combined with RoPE, Transformer XL style attention, Alibi, (and also with Flash Attention), etc. None of these require any modification to our CUDA kernel.

---

> ### Comment · Reviewer_obUv · 2023-11-17
> **Thanks**
>
> Thanks for the clarification.
>
> I appreciate the clarification. But it brings back the question. MACs are not a great metric. It depends heavily on the hardware.
>
> "Question regarding fine-tuning" :  I am asking a very simple question, to use your method on say a model we would need to first perform some amount extra training or not ?
>
> Despite all three reviewers pointing out how your method is different than the goal of flash attention, i find it interesting that you are not willing to change your position. You have a approximation based method which prunes the number of heads in transformer layer with gating (at a high level), flash attention concerns with build a hardware friendly implementation of exact attention operation. Two very different things in my opinion (of course you can disagree), which is why all reviewers have been pointing out that there is fundamental difference. "Research Friendliness" is a subjective metric. I find it very hard to digest the positioning of your method.

---

> > ### Author Response · Authors · 2023-11-18
> >
> > We would like to thank the reviewer for the prompt reply.
> >
> > > I appreciate the clarification. But it brings back the question. MACs are not a great metric. It depends heavily on the hardware.
> >
> > We respectfully note that this is the complete opposite: MACs are **hardware-independent** (as we also reminded in our previous response), and that is exactly their advantage over runtime that is heavily dependent on hardware and the implementation.
> >
> > (Just as a reminder: for example, the naive matrix multiplication algorithm for a square matrix of size N x N requires N^3 MAC operations independently of the hardware it is running on.)
> >
> > > "Question regarding fine-tuning" : I am asking a very simple question, to use your method on say a model we would need to first perform some amount extra training or not ?
> >
> >
> > We are sorry but there still seems to be some fundamental misunderstanding regarding our method. Our work is purely about model architecture. In our experiments, we train our models *from scratch*, in a single training phase (no pre-training or fine-tuning). So in this sense, no, we do not need to perform any “extra” training. Please note that this is the basic/standard setting (no fine-tuning) for papers proposing architectural changes see, e.g., [1,2,3].
> >
> > > You have a approximation based method which prunes the number of heads in transformer layer with gating (at a high level)
> >
> > This is not at all correct. Our method is **not** a pruning method.
> >
> > More generally, we really do not understand where these misunderstandings of the reviewer come from. Neither the word “pruning” nor “fine-tuning” is used in the paper.
> >
> > > flash attention concerns with build a hardware friendly implementation of exact attention operation. Two very different things in my opinion (of course you can disagree), which is why all reviewers have been pointing out that there is fundamental difference.
> >
> > This is exactly what we are claiming. They are fundamentally different. Even in the abstract of the paper, we write “The recently proposed Flash-Attention reduces both run-time and memory through a hardware-aware implementation. Can we achieve this also through algorithmic improvements?” highlighting exactly this difference.
> >
> > > Despite all three reviewers pointing out how your method is different than the goal of flash attention, i find it interesting that you are not willing to change your position. …
> > > "Research Friendliness" is a subjective metric. I find it very hard to digest the positioning of your method.
> >
> > Our claim on “research friendliness” is not at all subjective. As we also clarified in our response to Reviewer DLPx (explaining why our method is “research friendly” when we also need a CUDA kernel) and to Reviewer dyhc, any modification to the “attention mechanism” in FlashAttention requires modifications of their CUDA kernel (which requires proficiency in GPU programming): this is unquestionable. For example, you can not apply FlashAttention to Transformer XL, our baseline model (without changing the CUDA code): this is an indisputable fact. Nothing is subjective here. If the reviewer disagrees with this, we need a compelling explanation.
> >
> > Overall, we respectfully emphasize that we find it very difficult to follow the reviewer’s argumentations due to many fundamental misunderstandings (MACs, finetuning, pruning) as well as heavy reliance on other reviewer’s argumentations. It is absolutely fine that the reviewer refers to other reviewer’s concerns at this stage, but please do not ignore the corresponding responses we already provided (e.g., Reviewer dyhc on the subjectivity of research friendliness); If the reviewer disagrees with our arguments, we’ll be happy to respond, but it is not constructive to just repeat that the reviewer agrees with others while ignoring our response. We remind again that the reviewer is voting for clear rejection with a confidence score of 4. We’d appreciate constructive argumentations and justifications. Thank you.
> >
> > 1. Fedus et al, 2021: Switch Transformers: Scaling to Trillion Parameter Models with Simple and Efficient Sparsity
> > 2. Zhang et al, 2023: Mixture of Attention Heads: Selecting Attention Heads Per Token
> > 3. Dai et al, 2019: Transformer-XL: Attentive Language Models Beyond a Fixed-Length Context

---

> > > ### Comment · Reviewer_obUv · 2023-11-20
> > > **Thanks for the clarifications.**
> > >
> > > Thanks for the clarifications and my apologies, regarding misconstruing your method to be pruning method. I understand that you are proposing architectural changes rather than fine-tuning or pruning.
> > >
> > > I don't think we will agree with MAC's. In my experience reduction in MAC's does not mean gain in runtime.  A clear example is Sparse Networks like those generated by Lottery Ticket Hypothesis. They provide massive reduction in FLOPs but almost no benefit in runtime. Therefore optimizing for MACs I think is not fruitful.
> > >
> > > I understand what your method does now. Thanks for the clarification.  Do you think a more fair comparison would have been multi-query attention and grouped-query attention. [1]
> > >
> > > Now coming to the question of me referring to other reviewers arguments. This is a standard practice in any reviewing. We collaborate over different points, you make it sound that it is not acceptable to you (I do not agree with your response).
> > >
> > > [1] https://arxiv.org/abs/2305.13245
> > >
> > > Now based on some of you additional experiments showing runtime improve I am bumping the score. But I still think your paper needs a lot repositioning and additional comparison with say grouped-query attention etc. I also hope rather than being belligerent you understand the position and use the review as a vessel to improve the paper.

---

> > > > ### Author Response · Authors · 2023-11-20
> > > >
> > > > We would like to thank the reviewer for his response and his time.
> > > >
> > > > > I don't think we will agree with MAC's. In my experience reduction in MAC's does not mean gain in runtime. A clear example is Sparse Networks like those generated by Lottery Ticket Hypothesis. They provide massive reduction in FLOPs but almost no benefit in runtime. Therefore optimizing for MACs I think is not fruitful.
> > > >
> > > > We first would like to respectfully note that the reviewer is bringing up an argument that is completely different from the one in his/her previous response. Previously the reviewer stated “MACs are not a great metric. It depends heavily on the hardware.”, which is not true. Now the reviewer states that MACs do not not translate directly to wall-clock time acceleration. This is true, and we strongly agree with this. This is why we added wall-clock timing results to our appendix during our first update to the paper: see Table 7 in Appendix 4. **So we proved that our method provides wall-clock acceleration.** Given this, we do not understand what exactly the reviewer expects from us. **We report both MACs and runtime acceleration.**
> > > >
> > > > > I understand what your method does now. Thanks for the clarification. Do you think a more fair comparison would have been multi-query attention and grouped-query attention. [1]
> > > >
> > > > We would like to thank the reviewer for pointing out this interesting paper. We did not know about it until now. After checking it out we think it is slightly less relevant to compare against [1] compared to MoA. Here’s why:
> > > > [1] saves compute by reducing the number of linear projections used for keys and values, by sharding them between heads. The number of computed attention matrices stays the same as for the original, dense model. This provides speedup, but it is less than that of EPA or MoA. The reason being that MoA and EPA effectively do the same (shares some projections, in fact, the same projections as MoA), plus they do some additional compute reductions. Namely, they *significantly* reduce the number of the computed attention matrices, which is where most of the speedup and memory reduction comes from. They also reduce the number of (active) queries and output projections. Unfortunately, we cannot afford to run experiments with [1] for the review process because running a full set of experiments takes weeks of compute. We’d like to argue that we already provide comparisons to the most relevant baseline, MoA.
> > > >
> > > > > Now coming to the question of me referring to other reviewers arguments. This is a standard practice in any reviewing. We collaborate over different points, you make it sound that it is not acceptable to you (I do not agree with your response).
> > > >
> > > > We are not at all arguing against referring to other reviewer’s arguments (we wrote *“It is absolutely fine that the reviewer refers to other reviewer’s concerns at this stage”*). We argued against referring to the arguments of others without considering the responses we gave to them. Bringing up the same question again without stating why our provided answer is not satisfactory is not productive. Despite trying our best, it is impossible for us to address an issue if the reviewer does not clearly state why/what he/she disagrees with.
> > > >
> > > > > I do not agree with your response
> > > >
> > > > In the meantime, Reviewer dyhc clarified his/her concerns, and as a result, we removed all comparisons with Flash Attention from the paper. It is impossible for us to know what the reviewer refers to in “I do not agree with your response”, but we hope it was an argument against comparing with Flash Attention, which we hope we resolved now. Please see the updated PDF (unfortunately we cannot change the Abstract on OpenReview, so the modifications are present only in the PDF).
> > > >
> > > >
> > > > We hope that the current updates to the paper positions our work better and reflects what we originally intended to achieve. We hope this clears all the remaining doubts of the reviewer. If any remaining questions remain, we kindly ask the reviewer to state them clearly, and we will try to do our best to address them.

---

> > > > > ### Comment · Reviewer_obUv · 2023-11-20
> > > > > **Thanks**
> > > > >
> > > > > I have consistently held my position against MAC's.
> > > > >
> > > > > ```Regarding MAC operations and not runtime as a metric - We have time and time seen MAC operations are not a substitute for actual runtime operation, whether it be pruning, quantization or compression```. (First Comment) and you spent a bunch of time justifying MACs. It is on the insistence of the reviewers you updated your manuscript . Which I have acknowledged and based on that have updated the score.

---

> > > > > > ### Author Response · Authors · 2023-11-21
> > > > > >
> > > > > > Thank you very much for the score update.
> > > > > >
> > > > > > Now naturally, we'd like to know the reasons/justifications why the reviewer still votes for borderline reject. It seems that we thoroughly addressed and resolved all the main concerns the reviewer raised. We would also like to point out that reviewer dyhc increased the score to borderline accept. If the reviewer agrees with his/her points, please consider raising the score. If not, please provide us a comprehensive summary of what remaining issues we have to fix for acceptance. We would like to respond to any remaining concerns. Thank you.
> > > > > >
> > > > > > In the meantime we ran a small instance of [1], parameter-matched to Transformer XL with 47M params. We use 2 groups with 5 queries each, to replace the original 10 attention heads of the model. The results are as follows:
> > > > > >
> > > > > > | Model | sel. mode | $n_{heads}$ | #params | Perplexity | MACs | Mem (floats) |
> > > > > > | --- | --- | --- | --- | --- | --- | --- |
> > > > > > | Transformer XL | - | 10 | 47M | 12.31 | 453.4M | 3.5M |
> > > > > > | GQA | - | 10 | 47M | 12.31 | 473.7M | 3.3M |
> > > > > >
> > > > > > Timings:
> > > > > >
> > > > > > | Model | Size | ms/iteration | Rel. iter. time | RAM/GPU | Rel. Mem. | #GPUs | GPU type |
> > > > > > | --- | --- | --- | --- | --- | --- | --- | ---- |
> > > > > > | Transformer XL | 47M | 770ms/iter | 1.0 | 20G | 1.0 | 1 | RTX 3090 |
> > > > > > | GQA | 47M | 825ms/iter | 1.07 | 20G | 1.0 | 1 | RTX 3090 |
> > > > > >
> > > > > > As these tables show, GQA successfully reaches the perplexity of the baseline model but it is slightly more computationally demanding. Why is there no speedup for GQA? This is because in our experiments, **all models** are configured to have the same number of parameters for fair comparison (for all models, we achieve this by increasing $d_{head}$ until the parameter count matches that of the baseline). For GQA, this means that its $d_{head}$ is increased to compensate for its reduced key/value parameters (the resulting $d_{head}$ is 79 for GQA while it is 41 for the baseline); this makes the overall attention slower than the baseline.
> > > > > >
> > > > > > 1. Ainslie et al, 2023: GQA: Training Generalized Multi-Query Transformer Models from Multi-Head Checkpoints

---

### Official Review · Reviewer_DLPx · 2023-10-31

**Soundness:** 2 fair
**Presentation:** 3 good
**Contribution:** 3 good
**Rating:** 6
**Confidence:** 3

**Summary:**

The paper introduces a modification to the attention mechanism by incorporating a mixture of experts in both the source (K, Q) and destination (V, O) projections. This modification enables the selection of fewer active heads, thereby reducing computational and memory costs during both training and inference. The paper is based on the premise that not all attention heads are necessary for a given task. By utilizing an expert to select the required heads, it is possible to decrease computation and memory expenses. The effectiveness of this algorithm is demonstrated by comparing its accuracy to that of the dense counterpart and by visualizing the attention matrices.

**Strengths:**

1. The paper is well-written and effectively highlights the issues with the current attention architecture in terms of computational and memory demands.
2. The paper conducts experiments on various datasets and compares its results with existing baseline methods, including MOA.
3. The paper conducts a thorough analysis of attention maps to facilitate a qualitative study and comparisons with conventional attention matrices.

**Weaknesses:**

1. The paper refers to FlashAttention multiple times and compares against their CUDA kernel (SW designed to exploit HW efficiently) optimization vs algorithmic insight in this paper. I am not sure if its an apple-to-apple comparison since there are tons of other literature for transformers which aim to reduce computation/memory cost (like quantization/sparsity methods) and the paper doesn’t compare against these.
2. While authors compare against FlashAttention custom kernel implementation and mention that as a drawback, EPA algorithm itself requires a custom CUDA kernel with its own set of restrictions (pointed in the results section).
3. For the EPA algorithm, the paper mentions that K/Q source experts are not necessary for good results and only output/value experts are required, which seems to contradict the disadvantages shown in 2.2 naive algorithm.

**Questions:**

1. Can the authors compare against architectures other than TransformerXL? It is not evident from the text why only 1 architecture is chosen for comparison?
2. It is not evident from the paper how nHead is chosen for a task. Most results demonstrated fixed the nHead to be 2 or 4. Did the authors perform smaller experiments to first search for optimal nHead before scaling up?

---

> ### Author Response · Authors · 2023-11-15
>
> We would like to thank the reviewer for the insightful review, and for positive comments on the clarity and methodology of the paper. Please find our responses as follows:
>
> ## Clarification on the weaknesses
>
> > The paper refers to FlashAttention multiple times and compares against their CUDA kernel (SW designed to exploit HW efficiently) optimization vs algorithmic insight in this paper. I am not sure if its an apple-to-apple comparison since there are tons of other literature for transformers which aim to reduce computation/memory cost (like quantization/sparsity methods) and the paper doesn’t compare against these.
>
> We’d like to clarify one very important point regarding the goal of this work. The general goal of typical research on efficient attention is to make/train/evaluate Transformer models faster. Here we emphasize that such methods also have another important potential consequence in **accelerating research** on attention based models (by making them resource efficient). We consider this as an extremely important aspect today, potentially allowing for “resource-limited” organizations (e.g., in academia) to still conduct research on reasonably large models.
> However, for a “fast attention” method to be effective in accelerating research, it also has to be “research friendly”: certain existing methods do not allow modifications of certain aspects of attention mechanisms, while others may be more flexible. Such flexibility is required for “research friendliness”.
>
> This perspective is at the heart of our comparison with FlashAttention or with any other methods. We refer to FlashAttention simply as an example of a non-flexible acceleration method (and because it is arguably the most popular “fast attention” method today): it does not allow a researcher to modify the attention mechanism itself (which is hard-coded within their CUDA kernel), while our method still leaves such a possibility (since our CUDA kernel only replaces the linear projection layers to obtain query/key/values)---This should also bring an answer to the reviewer’s second point:
> > While authors compare against FlashAttention custom kernel implementation and mention that as a drawback, EPA algorithm itself requires a custom CUDA kernel with its own set of restrictions
>
> For example, FlashAttention can **not** be used with certain positional encoding (e.g. XL-style relative positional encoding) or activation functions (e.g. geometric attention). In contrast, our method can be trivially combined with RoPE, Transformer XL style attention, Alibi, (and also with Flash Attention), etc. None of these require any modification to our CUDA kernel.
>
> > For the EPA algorithm, the paper mentions that K/Q source experts are not necessary for good results and only output/value experts are required, which seems to contradict the disadvantages shown in 2.2 naive algorithm.
>
> We are not sure to understand what “contradiction” the reviewer refers to. Could you please clarify?
>
> ## Response to the questions
>
> > Can the authors compare against architectures other than TransformerXL? It is not evident from the text why only 1 architecture is chosen for comparison?
>
> We’d like to draw the reviewer’s attention to the fact that our experiments already include an architecture other than TransformerXL: standard (pre-layer norm) Transformer with RoPE positional encoding in Sec. 4, page 8.
> The main reason why we chose Transformer XL as our main architecture is that it has significantly better performance with a moderately low context window size (we use context windows of 256 for small models and 512 for big). For RoPE, we double the context window size of the XL transformers (512 for small, 1024 for big) to obtain competitive models (and even in this case, they underperform the XL models).
>
> > It is not evident from the paper how nHead is chosen for a task. Most results demonstrated fixed the nHead to be 2 or 4. Did the authors perform smaller experiments to first search for optimal nHead before scaling up?
>
> No, that is not how we chose $n_{head}$. Our search for $n_{head}$ works as follows: we start with $n_{head}=2$, and if the resulting model performs worse than the baseline, we double it to $n_{head}=4$. We never went beyond $n_{head} > 4$, as 2 or 4 was always enough to obtain good models. We added the corresponding explanation in the updated PDF in Appendix 2.1. Thank you for pointing this out.
>
>
> We believe we’ve addressed all the concerns raised by the reviewer. If you find our response convincing, please consider increasing the score. Otherwise, we’ll be happy to discuss and/or clarify further.

---

### Author Response · Authors · 2023-11-20
**Major update to the paper**

We updated our paper to remove all references to Flash Attention. Even though we intended to use FlashAttention merely as an example, they caused a lot of confusion, while these references are not at all crucial to highlight the core contributions of our work. Now we also emphasize the achieved wall-clock speedup more as our strength.

We would like to thank the reviewers once again for their time and for pointing out these confusing aspects and helping to make our paper better.

---

### Meta-Review · Area_Chair_eARu · 2023-12-14

**Metareview:**

The paper modifies the attention mechanism by incorporating a mixture of experts which enables the selection of fewer active heads, reducing computational and memory costs during both training and inference. The assumption is that not all attention heads are necessary for a given task. This is well known by the community, see e.g. [1] which builds upon the same assumption.

There has been considerable discussion between reviewers and the authors about the paper. The main weaknesses which have been identified are related to the inadequate comparison with FlashAttention, the metric used for efficiency (MAC), and the lack of comparison with other strong baselines (such as quantization/sparsity methods). While the authors were initially reluctant, they ended up doing major changes in the paper, removing FlashAttention and now reporting wall clock time. However, it is my opinion that several other weaknesses still persist, including a better comparison to competing methods. The overall positioning of the paper should be adjusted to reflect the major changes done by the authors in the rebuttal period. The major changes make me feel that this is no longer the "same" paper which was originally submitted. The paper would benefit from a better positioning, more rigorous comparison to competing methods, and a fresh round of reviews.

[1] Analyzing Multi-Head Self-Attention: Specialized Heads Do the Heavy Lifting, the Rest Can Be Pruned
Elena Voita, David Talbot, Fedor Moiseev, Rico Sennrich, Ivan Titov. ACL 2019.

**Justification For Why Not Higher Score:**

Already mentioned in the meta-review.

**Justification For Why Not Lower Score:**

N/A

---

### Decision · Program_Chairs · 2024-01-16

Reject